# OpenReview forum: "DialGuide: Aligning Dialogue Model Behavior with Developer Guidelines"
_EMNLP/2023/Conference — EMNLP 2023 Findings_

### Official Review · Reviewer_mumb · 2023-08-04

**Soundness:** 3

**Excitement:**

3: Ambivalent: It has merits (e.g., it reports state-of-the-art results, the idea is nice), but there are key weaknesses (e.g., it describes incremental work), and it can significantly benefit from another round of revision. However, I won't object to accepting it if my co-reviewers champion it.

**Paper Topic And Main Contributions:**

This paper presents Dialguide framework, consisting of three tasks, guideline retrieval, response generation and response entailment verification. Using this framework, dailogue models can generate proper responses following the given guidelines. This work starts from two existing datasets, Blended Skill Talk and ProsocialDialog, constructing the corresponding datasets Dialguide-BST and Dialguide-Safety. Fine-tuning on these datasets, the authors validate the helpfulness of guideline for dialogue response generation, showing promising experiment results.

**Questions For The Authors:**

- Why are you using InstructDial as a base model instead of better instruction-tuned models like Vicuna?
- Why do you use DialT0 in the task Response Entailment Verification, but DialBart0 in the task Response Generation?

**Reasons To Accept:**

- The use of guideline to guide the dialog model in generating responses is promising.
- The annotated dataset is valuable.

**Reasons To Reject:**

- As said in lines 176-179, I understand that there will not be both Task 2 and Task 3 in a single dialog round. I would suggest that task three be used as a test for task two so that you can take these tasks as a pipeline.
- Since you're using an instruction-tuned model, why not train the model with the mix of A B C tasks. It is recommended to try to accomplish all three tasks using a single model.
- Maybe some security should be considered, if some malicious guideline is set up, can this model defend against it?
- Why did some tasks use only the Dialguide-BST dataset?
- No experiment to validate the generalization of guidelines.

**Reproducibility:**

4: Could mostly reproduce the results, but there may be some variation because of sample variance or minor variations in their interpretation of the protocol or method.

**Reviewer Confidence:**

4: Quite sure. I tried to check the important points carefully. It's unlikely, though conceivable, that I missed something that should affect my ratings.

---

> ### Author Rebuttal · Authors · 2023-08-29
>
> We sincerely appreciate your positive feedback on our work, particularly your recognition of the promise in using guidelines to steer dialog model responses and the value of our annotated dataset.
>
> We address each of the reviewer’s comments and questions below.
>
> > *Regarding reviewer's suggestion - "I would suggest that task three be used as a test for task two so that you can take these tasks as a pipeline."*
>
> Thank you for the suggestion to use Task Three as a test for Task Two, thereby creating a pipeline of tasks. We agree that this could be an interesting approach for further validation and efficiency.
>
> > *Since you're using an instruction-tuned model, why not train the model with the mix of A B C tasks. It is recommended to try to accomplish all three tasks using a single model.*
>
> We appreciate your recommendation to train a single model for all three tasks. We agree that using a single model for multiple tasks is more practical and efficient, particularly in real-world applications. Your suggestion is in line with broader trends in multi-task learning and model efficiency. While we did explore multi-task learning in our preliminary experiments, we didn't find significant performance improvements, which is why we chose not to include it. Nonetheless, we acknowledge its potential for enhancing efficiency.
>
> > *Maybe some security should be considered, if some malicious guideline is set up, can this model defend against it?*
>
> We have discussed this point in the limitation and Ethics section. As a practical solution, apart from developing policies for writing guidelines and human oversight, one could use toxicity classifiers as an automatic solution.
>
> > *Why did some tasks use only the Dialguide-BST dataset?*
>
> For guideline entailment verification, we focused solely on the BST domain due to data collection costs and the promising preliminary results showing that a model trained on BST generalizes well to novel contexts and guidelines.
>
>
> > *No experiment to validate the generalization of guidelines.*
>
> In response to the concern about the generalization of guidelines, we did conduct an experiment in which we fine-tuned a model on the DIALGUIDE-SAFETY dataset and then evaluated it on the DIALGUIDE-BST dataset. As shown in the last row of Table 7, the model exhibited the ability to retrieve guidelines and generate responses that are coherent, diverse, and safe, while also entailing the guidelines. This experiment suggests that our guidelines have a degree of transferability across different domains, even those not explicitly trained on. In all of our results (Table 3-8), the test set consists of both guidelines and contexts the model has not seen during training, showing good generalization to novel contexts and guidelines.
>
> However, we acknowledge a potential limitation that you and we have both considered: while our framework is designed to be broadly applicable, the guidelines in our current dataset may not fully generalize to significantly distinct domains. In such cases, additional domain-specific guidelines would likely be needed to ensure optimal performance. Fortunately, our framework allows for the easy incorporation of new guidelines, and only a few additional guidelines can suffice for adaptation to new domains.
>
>
> > Question - *Why are you using InstructDial as a base model instead of better instruction-tuned models like Vicuna?*
>
> We used InstructDial as our base model because it is the most relevant open-sourced model that has been instruction-tuned on a large collection of dialogue tasks. InstructDial has state-of-the-art performance on a variety of dialogue tasks, which suggests that it is well-suited for our task of generating responses that follow guidelines.
>
> > Question - *Why do you use DialT0 in the task Response Entailment Verification, but DialBart0 in the task Response Generation?*
>
>
> We chose to use DialBart0 for the task of response generation because it has been shown to be as good as DialT0 at generation, while DialT0 is better at classification tasks. Additionally, text generation inference is much cheaper on the smaller DialBart0 model compared to the much larger DialT0 model.
>
>
> We hope that this addresses your concerns. Thank you again for the helpful suggestions and your feedback.

---

### Official Review · Reviewer_EDyR · 2023-08-05

**Soundness:** 3

**Excitement:**

4: Strong: This paper deepens the understanding of some phenomenon or lowers the barriers to an existing research direction.

**Paper Topic And Main Contributions:**

This paper proposes a new dialogue task called DIALGUIDE and the human-annotated dataset for the task. The task is to make a dialogue system response by following the guidelines provided. The task is divided into three subtasks - guideline selection, guideline-grounded response generation, and guideline-based response entailment verification tasks. For each subtask, there are several baselines are introduced, including OPT-30B few-shot and DIALGUIDE-finetuned T5/BART architecture models. The experiments suggest that fine-tuning a model on the DIALGUIDE dataset can make the model generate responses coherent, diverse, and safe.

**Questions For The Authors:**

1. My biggest question is the transferability of the dataset/task settings. It is not so clear how we can get benefits from this dataset.

2. Numbers on Table 4: although it seems models performing uniformly worse in the Adversarial test set, the performance drop is not so significant. (I'm comparing performance drops in the Adversarial NLI dataset [Nie et al., ACL2020]). Would it be feasible to make the test set even more challenging while maintaining the quality?

3. What is the length of the dialogue history for the input?

4. Which experiments correspond to the conclusion of that the model generalize well to new guidelines and contexts?

**Reasons To Accept:**

1. This paper proposes interesting task framing to realize a dialogue system in which we can control its dialogue flow based on tailor-made guidelines.

2. The paper is well-written and easy to follow. Overall, the experimental design is fair.

**Reasons To Reject:**

1. The transferability of this dataset is questionable. Since the task design is specific, it is not easy to transfer the knowledge acquired in this task to a more general type of dialogue system. Even if it is more for a specific design of dialogues, something that can inherit the way to generate a response in inference, the dataset may not be helpful since the covered domain of the dialogues is limited and may not overlap at all. On the other hand, since the task setting is different from the common dialogue response generation and requires some other abilities, performing badly on this dataset does not necessarily mean it is a bad dialogue system.

2. The reported results look decent enough, thus not sure whether recent models would seriously struggle with the task especially if they are fine-tuned on this dataset.

**Reproducibility:**

4: Could mostly reproduce the results, but there may be some variation because of sample variance or minor variations in their interpretation of the protocol or method.

**Reviewer Confidence:**

4: Quite sure. I tried to check the important points carefully. It's unlikely, though conceivable, that I missed something that should affect my ratings.

**Typos Grammar Style And Presentation Improvements:**

Figure 1 is a bit confusing because there are speaker 1/2 and not really sure it is intended for controlling which side of behavior (maybe better in user/system?)
I think all the utterances of a dialogue can be used to evaluate (at turn t, input the dialogue history up to turn t-1 and do all three tasks), but since the title sounds like you have a user/system conversation with the system side to control.

---

> ### Author Rebuttal · Authors · 2023-08-29
>
> We're very pleased that our paper resonated with you, particularly our approach to task framing and the clarity and fairness of our experimental design. We appreciate your feedback and will address your concerns in the following rebuttal.
>
> > Regarding Transferability
>
> Guidelines in our framework are expressed in natural language, leveraging the instruction-following capabilities of recent language models. This suggests that the guidelines should be transferable to other domains and tasks.
> To support this claim, we conducted an experiment in which we fine-tuned a model on the DIALGUIDE-SAFETY dataset and then evaluated it on the DIALGUIDE-BST dataset. The results (last row of Table 7) showed that the model was able to retrieve guidelines and generate responses that were coherent, diverse, entail the guidelines and are safe. This suggests that the guidelines are transferable to other domains, even if they are not explicitly trained on those domains. It's also worth mentioning that throughout our experiments (as shown in Tables 3-8), the test sets only contain unseen guidelines and contexts. The consistently strong performance of our models across these novel scenarios further validates their ability to generalize effectively.
>
> However, we acknowledge a potential limitation: while our framework is designed to be broadly applicable, the guidelines in our current dataset may not fully generalize to significantly distinct domains. In such cases, additional domain-specific guidelines would likely be needed to ensure optimal performance. Fortunately, our framework allows for the easy incorporation of new guidelines, and finetuning on only a few additional guidelines can suffice for adaptation to new domains.
>
>
> > *Regarding Task Specificity and General Dialogue Systems -  performing badly on this dataset does not necessarily mean it is a bad dialogue system.*
>
> It's important to note that our dataset and task design focus on specific capabilities, such as guideline adherence, which may not be the sole metrics for evaluating the efficacy of a general dialogue system. A model that underperforms on our dataset may still excel in other dialogue scenarios that require different skills or abilities. Our tasks aim to offer a focused evaluation on aspects like safety and guideline adherence, and should not be viewed as a comprehensive assessment of a dialogue system's overall quality.
>
> > *Regarding - The reported results look decent enough, thus not sure whether recent models would seriously struggle with the task especially if they are fine-tuned on this dataset.*
>
> We acknowledge that recent models are likely to exhibit strong performance. However, the value of our framework lies in its ability to further improve the alignment and safety of these already powerful models. By integrating them into our DialGuide framework, we can leverage their instruction following capabilities while ensuring that the generated responses are not only coherent and diverse but also adhere to specific guidelines for safety and context-appropriateness. Thus, our framework serves as an additional layer that enhances the utility of recent models in dialogue systems.
>
> > Regarding Adversarial Test Set Performance
>
> We'd like to clarify the naming and composition of our adversarial test set. In Table 1, the Dialogue-bst test set for entailment verification comprises 1507 positive, 2465 negative, and 990 adversarial negative instances. The normal test set is evaluated using the 1507 positive and 2465 negative responses, while the adversarial test set in Table 4 includes 1507 positive, and both the 2465 negative and the 990 adversarial negative responses. Unlike the adversarial test set in Nie et al., which only contains adversarial instances, ours is a mix of both normal and adversarial negatives. Consequently, the performance drop in our setting will be more pronounced if evaluated solely on the adversarial negatives. To avoid confusion, we will rename "adversarial" to "hard test set" in the final version. We want to add that we are using more recent language models such as T0 compared to Nie et al which used Roberta. These LMs are better at handling adversarial instances, which is why the performance drop would not be as significant.
>
> > Regarding Length of Dialogue History
>
> The dialogue history is truncated to a maximum of 1024 tokens (truncation from the left), achieved by concatenating previous utterances. We remove utterances which make the context go longer than 1024 tokens.
>
> > Regarding Generalisation Experiments
>
> In all of our results (Table 3-8), the test set consists of both guidelines and contexts the model has not seen during training. We show the performance of models and baselines on novel contexts and guidelines and the results show that our models generalize well to them. These results show the our models get above 90% coherence and safety, and around 90% guideline entailment score on novel guidelines and contexts. Furthermore, as we mentioned above, the last row in Table 7 shows the cross-domain transfer performance of our model, where a model trained on BST data performs well on safety domain contexts and guidelines.
>
>
>
> We hope that this addresses your concerns. Thank you again for your feedback.

---

### Official Review · Reviewer_QdKf · 2023-08-06

**Typos Grammar Style And Presentation Improvements:** NA
**Soundness:** 3

**Excitement:**

3: Ambivalent: It has merits (e.g., it reports state-of-the-art results, the idea is nice), but there are key weaknesses (e.g., it describes incremental work), and it can significantly benefit from another round of revision. However, I won't object to accepting it if my co-reviewers champion it.

**Missing References:**

NA

**Paper Topic And Main Contributions:**

In order to generate engaging and safe responses, this paper proposes a novel framework for controlling dialogue model behavior using natural language rules.

They proposed three sub-tasks in the framework, (1) Guideline selection; (2) Response generation; (3) Response entailment verification.

**Questions For The Authors:**

See Reasons To Reject

**Reasons To Accept:**

This paper benchmarks three tasks for prompt-enhanced dialog generation.

**Reasons To Reject:**

(1) Using prompts to improve model performance is a common widely adopted practices. I'm not sure about the contribution of this paper, especially when they don't promise to open-source the labeled data.


**Reproducibility:**

3: Could reproduce the results with some difficulty. The settings of parameters are underspecified or subjectively determined; the training/evaluation data are not widely available.

**Reviewer Confidence:**

4: Quite sure. I tried to check the important points carefully. It's unlikely, though conceivable, that I missed something that should affect my ratings.

---

> ### Author Rebuttal · Authors · 2023-08-29
>
> We sincerely appreciate your time and effort in reviewing our paper. Your feedback is invaluable, and we would like to address the concerns you raised.
>
> > *Regarding the availability of code and data*
>
> We apologize for any confusion regarding the availability of code and data. We will release the code and data for our paper upon publication. We have mentioned this in the footnote on page 2, and we will add a link in the final version of the paper.
>
>
> > *Regarding the use of prompts and our contributions*
>
> We agree that prompting-based approaches have gained widespread acceptance within the NLP community, and our work, DialGuide, certainly builds upon this well-established area of research. However, DialGuide elevates the concept beyond using static prompts for enhancing model performance. While traditional methods typically employ a limited set of static prompts across varying contexts, DialGuide is engineered to dynamically fetch the most relevant and context-appropriate guidelines and use them as prompt. This adaptability addresses a significant limitation of traditional prompting techniques, which often struggle to adapt to complex, ever-changing scenarios. In stark contrast, DialGuide's adaptive prompting mechanism facilitates more nuanced, controlled, and coherent responses, uniquely tailored to specific contexts.
>
> Beyond these points, we offer the following unique contributions that differentiate our work:
> * Novel Framework and Data: We introduce DialGuide, a novel framework designed to control dialogue model behaviour using natural language rules or guidelines. This is not merely an application of existing prompting techniques but a comprehensive framework that aims to solve a more complex problem.
> * Modelling Contributions: Our work decomposes the task into three distinct sub-tasks:
>     1. Guideline Retrieval: Identifying the most appropriate guidelines relevant to the context.
>     2. Response Generation: Creating a response that adheres to the selected guideline.
>     3. Response Entailment Verification: Assessing whether the generated response is in alignment with the guideline.
> * Benchmark and Results: We establish a robust set of baselines and benchmark performance on these tasks. Our results demonstrate that models trained on our dataset can generate more controlled and coherent responses.
>
>
> We hope that this addresses your concerns. Thank you again for your feedback.

---

### Meta-Review · Area_Chair_Drob · 2023-10-06

**Recommendation:** 3

**Metareview:**

The authors introduce a new dialogue task called DIALGUIDE and provide a human-annotated dataset for this task. The experiments conducted in the paper demonstrate that fine-tuning a model on the DIALGUIDE dataset can improve the model's ability to generate coherent, diverse, and safe responses.  The experimental results indicate promising outcomes in terms of response quality and safety.

---

### Decision · Program_Chairs · 2023-10-07

**Decision:**

Accept-Findings

**Comment:**

The authors introduce a new dialogue task called DIALGUIDE and provide a human-annotated dataset for this task. The experiments conducted in the paper demonstrate that fine-tuning a model on the DIALGUIDE dataset can improve the model's ability to generate coherent, diverse, and safe responses.  The experimental results indicate promising outcomes in terms of response quality and safety.